# Estimation Formula of Modal Frequency of High-Rise Buildings under Different Wind Speeds during Typhoons

**Jiaxing Hu [1,\*], Zhengnong Li [2] and Zhefei Zhao [3]**

[1] School of Civil and Environmental Engineering, Hunan University of Science and Engineering, Yongzhou 425199, China

[2] Key Laboratory of Building Safety and Energy Efficiency of the Ministry of Education, College of Civil Engineering, Hunan University, Changsha 410082, China; zhn88@263.net

[3] School of Vocational Engineering, Health and Sciences, RMIT University, GPO Box 2476, Melbourne, VIC 3001, Australia; zhefeizhao@gmail.com

\* Correspondence: wjxcivil@yeah.net

**Abstract:** On 18 October 2016, the wind-induced effects of a high-rise building with square section was measured by the monitoring system in Haikou of China during Typhoon Sarika. The wind characteristics atop the building and the time-history responses of the translational and rotational accelerations on different floors were measured by the monitoring system; the first three modal parameters were identified according to the measured acceleration. The results show that the combinations of the cross spectral density function, phase spectrum, and coherence function can clearly judge the phase of the measured floors in the frequency resonance area as well as its modal frequencies at the first three orders. The modal frequencies at the first three orders decrease linearly with the growth of mean wind speed within the range of 0~20m/s. The estimation formula of the modal frequencies of high-rise buildings considering the influences of different wind speeds is put forward, which is expected to fill the gap in the existing specification for the quantitative analysis of the influences of wind-loads on the fundamental frequencies of high-rise buildings.

**Keywords:** typhoon; high-rise buildings; wind-induced responses; modal parameter; fundamental frequency

## 1. Introduction

According to the measured vibration signal data, the modal parameter identification of high-rise buildings is the processing method used to estimate the natural frequency, damping ratio, and vibration mode of the structure. With the characteristics of light weight, high flexibility, and low damping, modern high-rise buildings are more sensitive to wind-induced vibration, thus the reliable estimations of modal parameters are vital to design and evaluate the safety and applicability of structures. Likewise, accurate modal parameter information can be used to determine the safe running state, dynamic response behavior, material damage, and vibration control of structures. A three-dimensional accelerometer is widely used in the modal identifications of high-rise buildings, by which the measured acceleration is often regarded as the translation along the east–west, south–north, and vertical directions. However, the dynamic characteristics of all particles cover rotation besides translation, thus the vibration of structures on six degrees of freedom should be measured. Namely, the vibration conditions of structures can be systematically represented only by three rotations and three translations. A large number of studies have confirmed that the rotational component contributes a lot to the earthquake and wind-induced responses of structures. Further, many studies also showed that the torsional vibration components of high-rise buildings contribute a lot to the structural responses. Rotational components can produce giant shear and moment, and tall and slender buildings

are more inclined to torsional failure under the action of typhoons. However, because of some limitations of the measuring sensors, the rotational measurements are not as widely used as the translational measurements (acceleration, speed, and displacement).

The existing research mainly covers the torsion-resistant studies of structures under the actions of controllable load, earthquakes, and wind load. As for the torsion-resistant studies of the reinforced concrete structures under the action of controllable load, many scholars (Patil, 2016 [1]; Ahmed, 2014 [2]; Ren, 2011 [3]; Schladitz, 2012 [4]; Mohamed, 2016 [5]; Mohammed, 2015 [6]) conducted theoretical and experimental research on both reinforced concrete and steel-concrete composite structures, and acquired a series of achievements. Many scholars (such as Hejal, 1989 [7,8]; Hsu, 2000 [9]; Siah, 2003 [10]; De-La-Colina, 2004 [11]; Hsu, 2010 [12]; Balkaya, 2003 [13]; Basu, 2006 [14]; LV, 2002 [15]; XU, 2000 [16] ; HE, 2002 [17]; PAN, 2004 [18]; Li, et al., 2006 and 2010 [19,20]) have studied the lateral-torsional coupling issues of structures under the actions of earthquakes by the theoretical analysis, finite element model, and vibration tests, and explored the influences of eccentricity $e/r$, period ratio $T_t/T_l$, and the relative torsional response ratio $\theta_r/u$ on structures. They also suggested that the torsional responses of eccentricity $e/r$ and the period ratio $T_t/T_l$ and relative torsional response ratio $\theta_r/u$ to structures should be controlled. While the existing field measurements to the torsional responses of high-rise buildings under the actions of earthquakes were insufficient, which was closely related to the factors of earthquakes such as unpredictability, burstiness, and great destructiveness, these factors also posed a challenge to conducting field measurements of earthquake resistance on high-rise buildings. Studies on the dynamic parameters of high-rise buildings under the influences of typhoons have been conducted by many scholars in recent years, but most involved the translational dynamic parameter identifications of high-rise buildings (Campbell, 2005 [21]; Li, 2017 [22], 2014 [23], 2004 [24]). Many existing torsion resistance studies analyzed the torsional wind load and the wind-induced responses of structures on the basis of wind tunnel tests and the related theories, while few studies were specially related to the prototype filed measurements of the torsional modal parameters of high-rise buildings under the influences of typhoons. Feng, et al. (2013) [25] studied wind-induced torsion vibration of the super high-rise building of Shenzhen Energy Center, and proposed that the wind-induced torsion vibration of the building was sensitive to wind directions, thus it should be considered. On the basis of processing and analyzing the measured data of the fluctuating pressures on the rigid rectangular cylinder models with various side ratios and aspect ratios in a boundary layer wind tunnel, Liang, et al. (2004) [26] presented the empirical formula for the power spectral densities of wind-induced dynamic torque, RMS torque coefficients, and Strouhal numbers for the isolated rectangular tall buildings with various side ratios. Yi, et al. (2017) [27] studied the interference effects of the torsional moments between two high-rise buildings on the basis of pressure and flow field measurement. By applying the rigid model wind tunnel test, Yu, et al. (2016) [28] analyzed the distribution variation and the correlation of envelope interference factor (EIF) of the base torsion responses for the principal building.

To sum up, the torsion responses of the structural components are complicated, especially of those high-rise buildings under the effects of typhoons. Nowadays, the studies on the torsion effects of structure components are mainly based on laboratory tests, theoretical analysis, finite element simulation, and seismic response measurements, while prototype tests on the torsional effects of high-rise buildings under the effects of typhoons are rarely studied. Therefore, there are practical and theoretical values to investigate the torsional responses of high-rise buildings under the influences of typhoons. By carrying out field measurements on the high-rise building with rectangular shape in Haikou, the translational accelerations on the 6th, 12th, 18th, 24th, 30th, and 32th floors and the torsional angular accelerations on the 8th, 12th, 16th, 20th, 24th, 28th, and 32th floors are obtained; and the translational and torsional modal frequencies, vibration modes, and the damping ratios at the first three orders are identified. This paper took the influences of wind-induced response on fundamental frequencies into consideration during typhoons, and

then the estimation formula of the modal frequencies of high-rise buildings considering the influences of different wind speeds was put forward.

## 2. General Introductions to Field Measurements

### 2.1. Landing of Typhoon Sarika

Typhoon Sarika landed in Wanning city of Hainan province at 09:50 on 18 October 2016. The maximum wind grade reaches 14 (45m/s) near the typhoon centre in the passage of Sarika, the minimum central pressure is 955 hPa, and the maximum wind affected Hainan appears around 10:00. The moving track of typhoon Sarika is presented in Figure 1. The 32-floor measured building (the view chamber locates at top floor) supported by four huge columns is a relatively high building near the coastline. The height of the top floor is 108 m, the aspect ratio is 6.71, and the width/thickness ratio is 1.47; more details are shown in Figure 1(a and b). Figure 2 shows the building plan of the measured high-rise building. The geographical coordinate of the high-rise building along the north main axis is 11° north by west. There is no refuge layer and structural reinforcement layer in the middle floor, and the mass and stiffness are evenly distributed along the height direction.

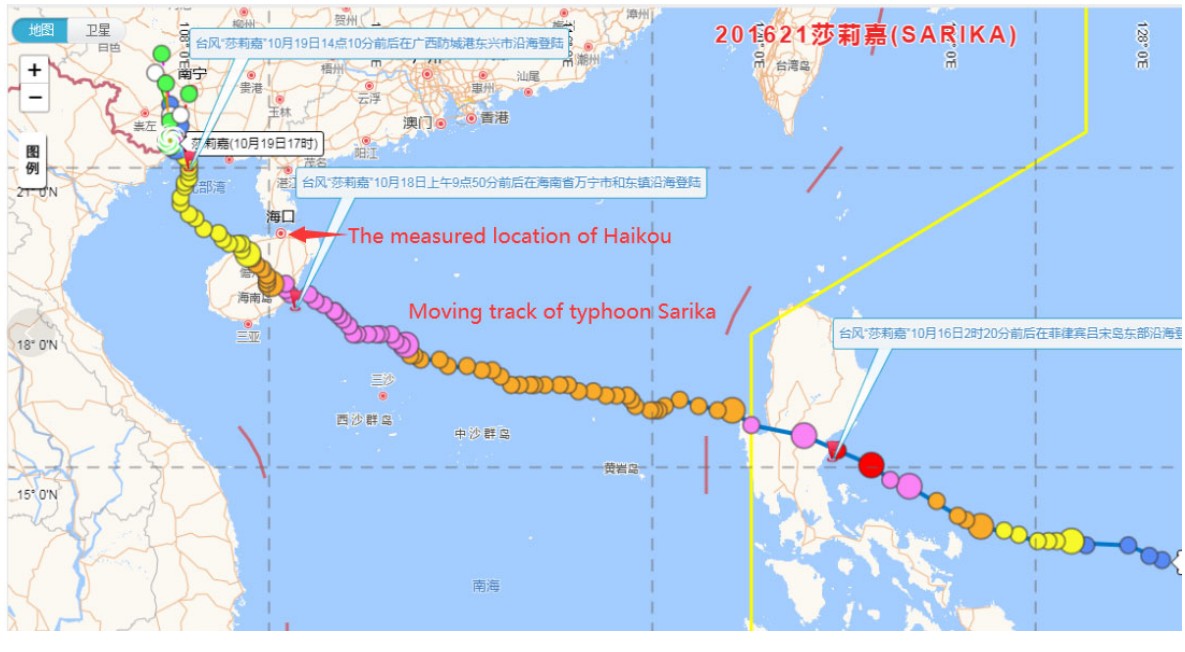

(**a**)

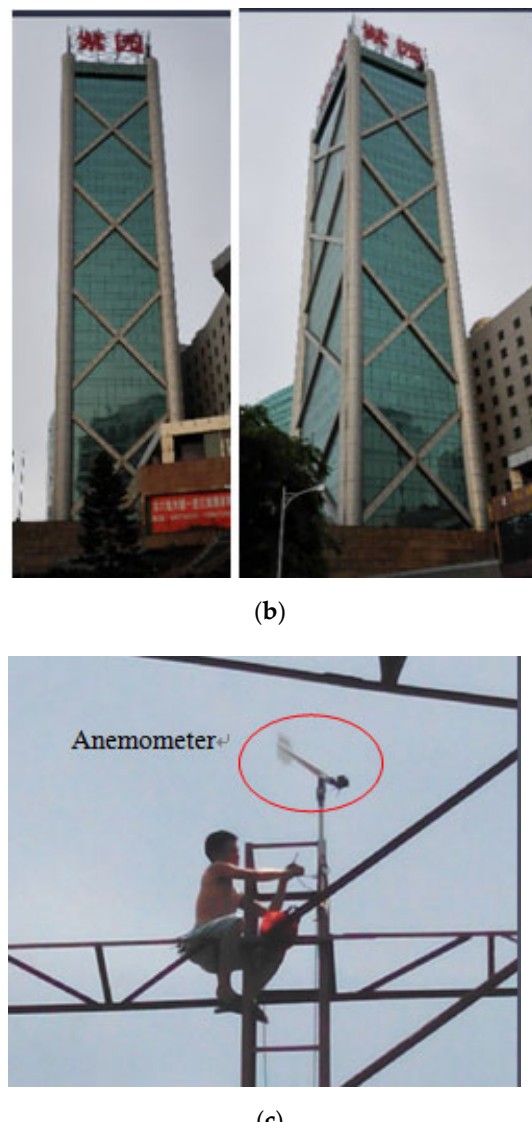

(**b**)

(**c**)

**Figure 1.** Moving track of typhoon Sarika and the exterior view of the measured building: (**a**) moving track of typhoon Sarika; (**b**) exterior view of the measured building; and (**c**) field installation of anemometer atop the building.

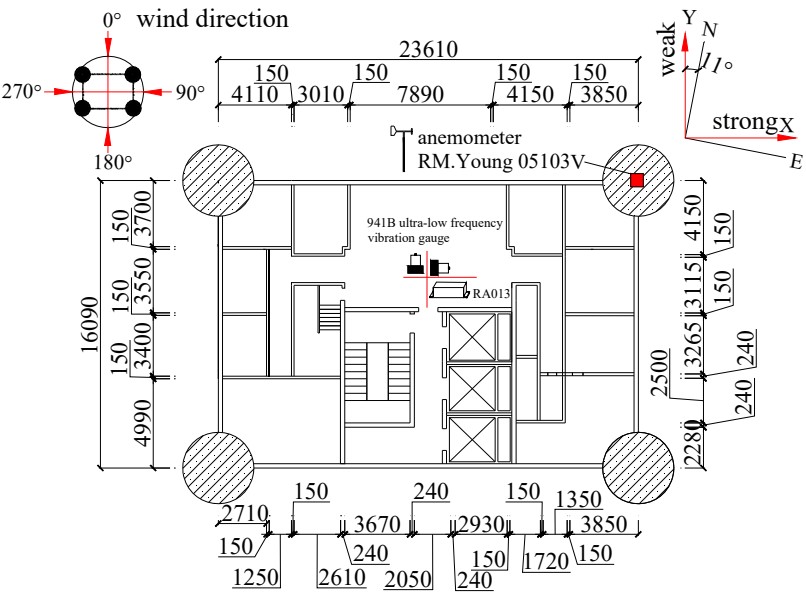

**Figure 2.** Building plan of the measured high-rise building.

## 2.2. The Monitoring System

The wind speed, wind direction angle, and the translational acceleration were all collected by the monitoring system produced by Wuhan Yutek Electronic Technology Co., Ltd. Model RM.Young 05103V mechanical anemometer was used to collect the wind field and was installed at the vertical weight of 115 m. Figure 1(c) presents the installation of anemometer on the top floor of high-rise building. The translational acceleration was monitored by model 941B ultra-low frequency vibration gauge. The torsional angular accelerations were measured by a model RA013 rotation accelerometer, which is a sensor used to measure the torsional angular accelerometer in engineering. The model RA013 rotation accelerometer has the favorable linear, low frequency performances, and large dynamic measurement range, and its parameters are as follows: the pass band is 0.01~20 Hz, the resolution is lower than $2 \times 10^{-4}$ rad/s$^2$, the size is 330 mm × 130 mm × 105 mm, and the height is 6 kg. The rotation sensitive axis of the rotation accelerometer is presented in Figure 3. The Z axis perpendicular to the bottom plate of rotation accelerometer is the measured rotation sensitive axis, and $s^2\theta$ refers to the ground rotation acceleration. Therefore, the installation shown in Figure 3 is used to measure the torsional angular acceleration around the Z axis. If the torsional angular acceleration around the X axis is necessary, then the rotation accelerometer should be turned inward by 90 degrees (the output plug should be put upward). Figure 4 shows the measuring point arrangements of the translation and rotation accelerometer and Figure 5 is the schematic diagram of the translation and rotation accelerometer arrangements on different floors.

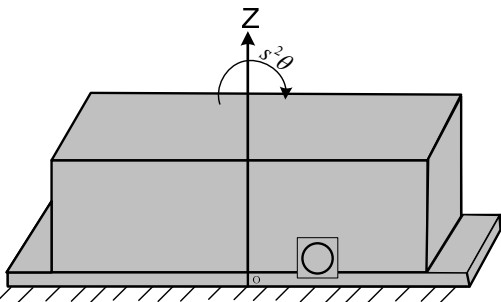

**Figure 3.** Rotation sensitive axis of the rotation accelerometer.

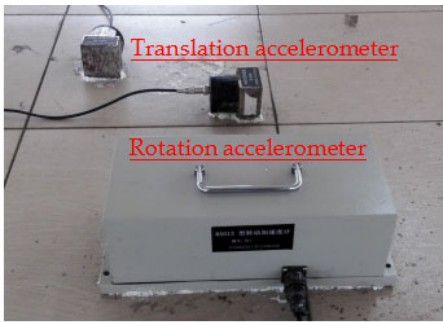

**Figure 4.** Measuring point layouts of the translation and rotation accelerometer.

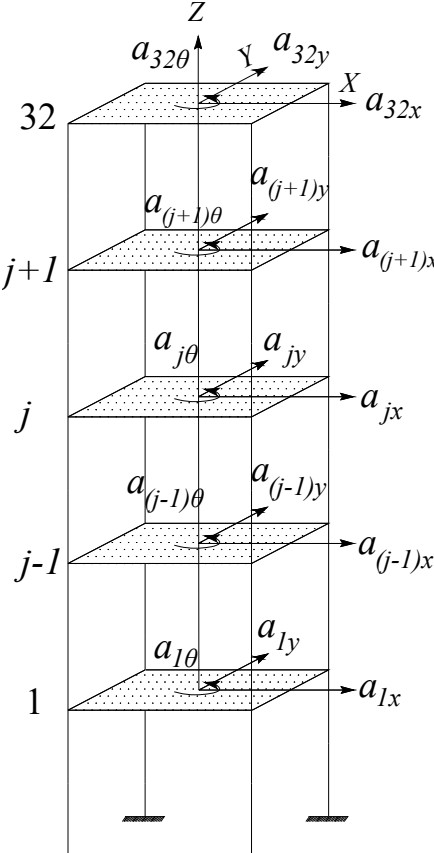

**Figure 5.** Layout of the translation and rotation accelerometers on different floors: X axis; Y axis; Z rotation axis.

### 2.3. Testing Principle of the Rotation Accelerometer

The mathematical representation of the mathematical model of the rotation accelerometer is shown in Figure 6. The motion differential equation of model RA013 rotation accelerometer can be expressed as follows (Yang, et al., 2015 [29]):

$$K_1\ddot{\theta} + b\dot{\theta} + k\theta + G_1 i_1 = -\sum m_i H_i \times \ddot{y} - K_1\ddot{\varphi} \tag{1}$$

where $K_1$ is the moment of the inertia of the double pendulum swinging about the rotational axis. $b$ refers to the damping coefficient. $k$ denotes the angular stiffness. $\theta$ represents the displacement. $\ddot{y}$ is the ground translation acceleration. $\ddot{\varphi}$ is the ground

rotational acceleration. $G_1$ is the electric constant of the damping coil. $i_1$ is the current flowing into the damping coil.

The solution of the equation can be expressed by (Yang, et al., 2015 [29])

$$\theta(s) = \frac{-s^2}{(s^2 + 2Dns + n^2)}(\frac{1}{l_0}y - \varphi) \tag{2}$$

where $l_0$ is the equivalent pendulum length. $s$ is the operator. $n$ is the circular frequency of self-vibration. $D$ is the damping ratio.

Furthermore,

$$n^2 = \frac{k}{K_1}, \quad 2Dn = \frac{b}{K_1}, \quad l_0 = \frac{K_1}{\sum m_i H_i} = \frac{K_1}{m_1 H_1 - m_2 H_2}$$

where $m_1, m_2$ are the effective masses of the double pendulum. $H_1, H_2$ are the distances between the two centers of mass and the rotational axis. Likewise, when $m_1 H_1 = m_2 H_2$, $l_0 = \infty$, the instrument will only reflect the rotational angle, and not the translation; the solution of the equation can be expressed by (Yang, et al. 2015)

$$\theta(s) = -\frac{s^2\varphi}{(s^2 + 2Dns + n^2)} \tag{3}$$

It can be seen from Equation (3) that the angle of rotation accelerometer is in proportion to the rotation accelerometer of ground $s^2\varphi$.

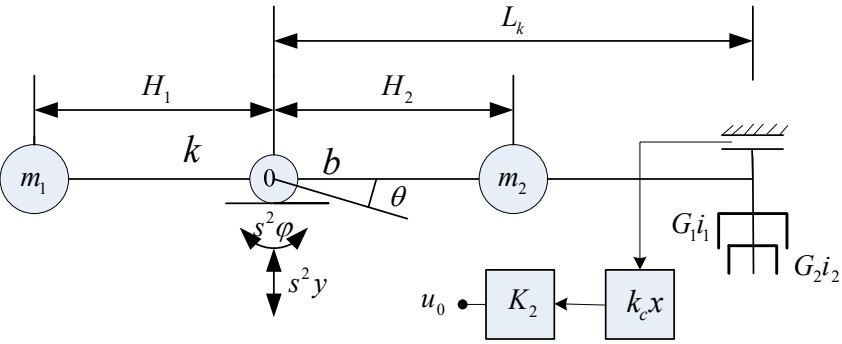

**Figure 6.** Schematic representation of the mathematical model of the rotation accelerometer (Yang, et al., 2015).

## 3. Measured Results Analysis

### 3.1. Time History of Wind Field Atop the Building

The measured data of wind field atop the building recorded within 36.5 h after 23:00 on 17 October 2016 were selected. The data of wind field illustrated the wind speed and wind direction angle atop the building in Haikou during the whole passage of typhoon Sarika, and the measured maximum instantaneous wind speed was about 34.84m/s. The measured inflow wind direction angle was basically from 0° to 180°, and the measured maximum mean wind speed was about 20.02 m/s when the basic time duration was defined as 10 min. More details are shown in Figure 7.

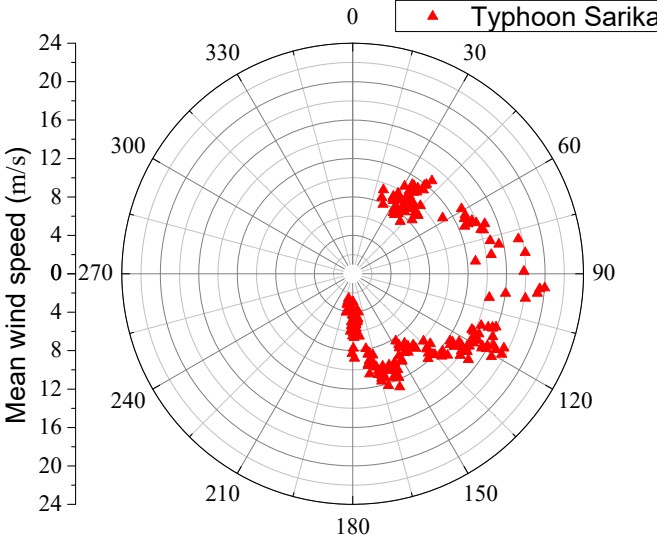

**Figure 7.** Wind field characteristics of typhoon Sarika.

### 3.2. The Measured Time History of Acceleration

As shown in Figure 5, the translation accelerometers were installed at the 6th, 12th, 18th, 24th, 30th, and 32th floors to obtain the translational acceleration along X and Y axes on different floors of the high-rise building under the influences of typhoon Sarika, by which the translational acceleration responses of six floors were recorded. The partial data of acceleration responses within 4.5 h after the landing of Sarika were selected. The time histories of the translational acceleration on six floors (6th, 12th, 18th, 24th, 30th, and 32th) are shown in Figure 8.

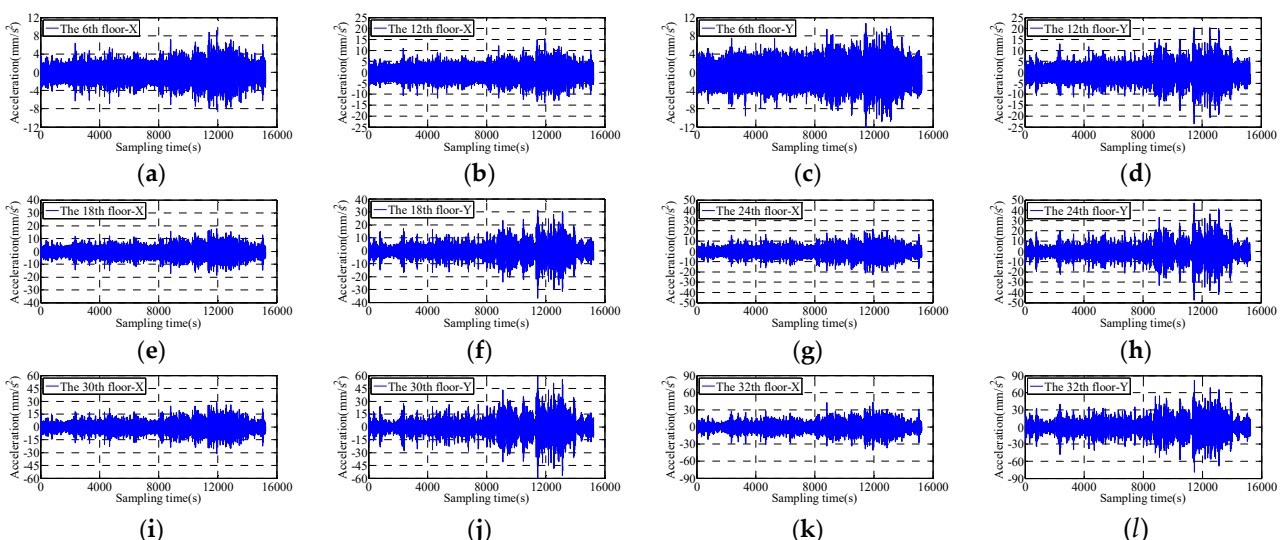

**Figure 8.** Translational acceleration time histories of Typhoon Sarika. (**a**) 6th-X axis;(**b**) 6th-Yaxis; (**c**) 12th-X axis;(**d**) 12th-Yaxis; (**e**) 18th-X axis;(**f**) 18th-Yaxis; (**g**) 24th-X axis;(**h**) 24th-Yaxis; (**i**) 30th-X axis;(**j**) 30th-Yaxis; (**k**) 32th-X axis;(**l**) 32th-Yaxis.

It can be known from Figure 8 that the translational acceleration responses at different floors along both the X and Y axes grow as the height of floors increases. The maximum acceleration, which appears on the 32th floor along the Y axis, was 0.079 m/s². Be-

sides, the time histories of translational acceleration root mean square and their peak values on the 6th, 12th, 18th, 24th, 30th, and 32th floors were also acquired when the basic time duration was 10 min. The results show that the peak values of accelerations and their root mean square all grow with the increasing floor height, and their development trends were essentially in agreement with each other.

The measured vibration modes of high-rise buildings can be obtained by synchronous measurement of acceleration response of different floors or moving measuring point method. In this measurement, sensors are arranged in four floors, and the angular acceleration synchronous response of four floors (8th, 16th, 24th and 32th) can be obtained. However, in order to obtain more floor vibration mode vectors, the moving measuring point method is adopted to identify the vibration modes of high-rise buildings, The moving measuring point method only needs two acceleration sensors (such as the acceleration sensors of 24th and 32th floors) to obtain the measured vibration modes of high-rise buildings. Therefore, the measured position of the sensor on the 32th floor can be set to be fixed, and the measured position of the other sensor can be continuously moved to obtain the synchronously measured acceleration of the 8th, 12th, 16th, 20th, 24th and 28th floors relative to the 32th floor respectively. The time histories of the angular accelerations on the 24th and 32th floors during Typhoon Sarika are presented in Figure 9, the maximum torsional angular acceleration is 0.031rad/s².

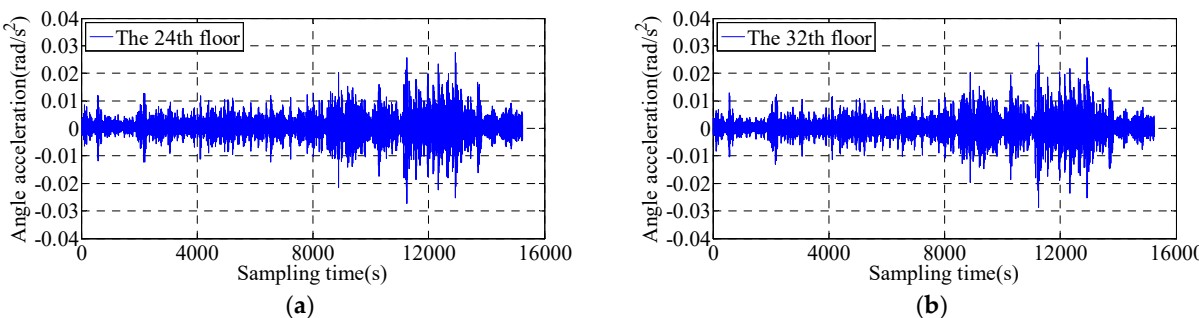

**Figure 9.** Time histories of the angular acceleration under typhoon Sarika.(**a**) Torsional angular acceleration response of 24th floor; (**b**) Torsional angular acceleration response of 32th floor.

### 3.3. Power Spectral Density Function

Auto-power spectral density (APSD) and cross-spectral density (CPD) are key characteristics of signal samples. The power spectrum, which enables to reflect the average characteristics of statistics, is the important statistical parameter to investigate the frequency domain characteristics of random vibration. The calculation formulas of the auto-power spectral and cross-spectral density functions can be expressed by Equations (4) and (5).

$$S_{xx}(k) = \frac{1}{MN_{FFT}} \sum_{i=1}^{M} X_i(k)X_i^*(k) \tag{4}$$

$$S_{xy}(k) = \frac{1}{MN_{FFT}} \sum_{j=1}^{M} X_j(k)X_j^*(k) \tag{5}$$

where $X_i(k)$ is the Fourier transform of the one-dimensional random vibration signal of a measuring point at the *i*th data segment; $X_j(k)$ is the Fourier transform of two-dimensional random vibration signal of a measuring point at the *j*th data segment; $X_i^*(k)$ is the conjugate complex of $X_i(k)$; $X_j^*(k)$ is the conjugate complex of $X_j(k)$; $N_{FFT}$ is the data length of Fourier transform; and *M* is the average time. The average periodic diagrams were used in this paper to estimate the power spectrum density, by

which the vibration signals were divided into several components, and the power spectrum of each component was calculated and then the average value was selected.

The time histories of the torsional acceleration on different floors are measured at the same time segment during typhoon Sarika, and the torsional angular acceleration responses of other floors are measured at different time segments, except the 24th and 32th floors, during typhoon Sarika. In order to obtain the torsional mode, the measuring points of the rotational accelerometer on the 32th floor were taken as the reference floor, and the rotational accelerometer was installed at the 24th floor, thus the angular acceleration responses of the 8th, 12th, 16th, 20th, 24th, and 28th floors that were synchronous with those of the 32th floor were obtained as well. Even though the power spectrum amplitudes in the period of relatively high wind speed are large, they do not affect the judgment to the fundamental frequencies of the high-rise building. According to Equation (4), the auto-power spectrum density of acceleration along the X axis and Y axis and the torsional direction were calculated. More details are shown in Figure 10.

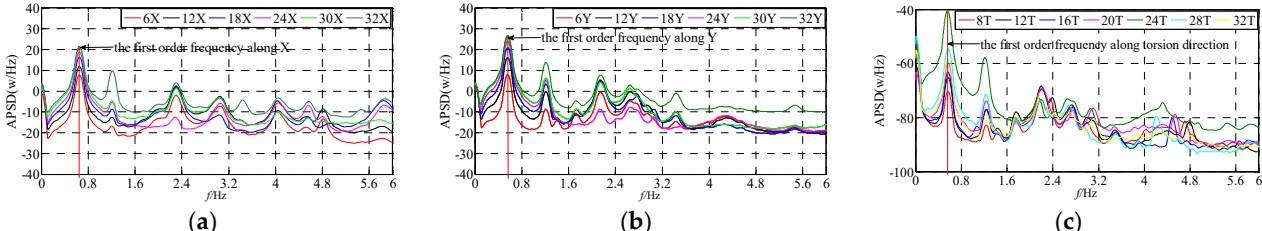

**Figure 10.** Acceleration auto-power spectral density functions: (**a**) auto-power spectral density along the X axis; (**b**) auto-power spectral density along the Y axis; (**c**) auto-power spectral density along the torsional direction.

It can be known from Figure 10 that the auto-power spectral density functions along X and Y axes and torsional direction have obvious peaks at the first-order frequencies, and the values of the first-order frequencies are obviously larger than those of the higher ones. However, the peak amplitude of the 24th floor at the first-order frequency along the torsional direction was almost the same as that of the 32th floor. The fundamental frequencies along the X and Y axes were about 0.66 Hz and 0.56 Hz, respectively, while the fundamental frequencies along the torsional direction were about 0.56 Hz. Therefore, the fundamental frequency along the torsional direction was close to that along the Y axis of the measured building. It shows that the first-order torsion-translation period ratios $T_t/T_l$ and $T_t/T_s$ are 1.18 and 1.00, respectively.

Because the wind energy is concentrated in the low-frequency region, it has a great impact on the measured low-order modal frequency of high-rise buildings. The measured results show that the frequency value at the self-spectral peak is significantly less than that at the low wind speed peak in the horizontal and torsional turning high-speed stage, indicating that the influence of the vibration amplitude of high-rise buildings on the horizontal and torsional turning modal parameters cannot be ignored.

*3.4. Cross Spectral Density Function (CSD), Phase Spectrum (PS), and Coherence Function (CF)*

In the frequency response function, frequencies corresponding with the peak value of the amplitude–frequency curve can be determined as the natural frequency, and the phase spectrum can reflect the specific phase and the phase-difference values of different floors in the resonance region. Taking frequency response as an example:

$$\mathrm{H}_d\left(\omega\right)=\left|\mathrm{H}_d\left(\omega\right)\right|\exp(j\varphi(w)) \tag{6}$$

$$\left|\mathrm{H}_d\left(\omega\right)\right| = \frac{1}{m\sqrt{\left(\omega_0^2 - \omega\right)^2 + \left(2\zeta\omega_0\omega\right)^2}} \tag{7}$$

$$\varphi\left(\omega\right) = \arctan\frac{-2\zeta\omega_0\omega}{\omega_0^2 - \omega^2} \tag{8}$$

where $\mathrm{H}_d\left(\omega\right)$ is frequency response function; $\varphi\left(\omega\right)$ is the phase-frequency function; $\omega$ is the circular frequency, rad/s; $m$ is mass, kg; and $\omega_0$ is the natural circular frequency of the structure.

Coherence function is the index of correlation degree for the two-dimensional random vibration signal in the frequency domain. The coherence function is the quotient obtained by dividing the square of the mode of the cross power spectral density function by the self spectral product of the excitation and response. Equation (9) is the calculation equation of the coherence function:

$$C_{xy}(k) = \frac{\left|S_{xy}(k)\right|^2}{S_{xx}(k)S_{yy}(k)} \tag{9}$$

where $S_{xx}(k)$ and $S_{yy}(k)$ are the estimations of the auto-power spectrum density function of random vibration excitation and response signals processed by the average periodic diagrams; and $S_{xy}(k)$ is the estimations of the cross-spectral density function of excitation and response signals. The coherence function ($C_{xy}(k)$) was adopted in this paper to evaluate the frequency response function. If $C_{xy}(k)$ became closer to 1, this meant that the noise disturbance was slight and the estimation results of frequency response function were good. The measured environment was complicated, because several factors, such as typhoon, normal wind, earth pulsation, mechanical vibration, and ambient noise, would affect the accuracy of the excitation function. Therefore, the cross-spectral density curves between the acceleration data on the 6th, 12th, 18th, 24th, and 30th floors along the X and Y axes and those of the 32th floors were presented, and the cross-spectral density curves between the torsional angular acceleration data of the 8th, 12th, 16th, 20th, 24th, and 28th floors and those of the 32th floor were also given, by which the corresponding phase spectrum and the coherence functions were sketched, and the cross-spectral density function, phase spectrum, and coherence functions along X axis and Y axis and the torsional direction were all obtained. More details are presented in Figures 11–13.

Figure 11 shows that wave crests of the cross-spectral density function between the 6th, 12th, 18th, 24th, and 30th floors and the 32th floors along the X axis are clearly visible, and the phase angles of all cross-spectral density function are 180° when the first-order frequency of wave crest is 0.66 Hz, which explains that the acceleration vibration directions of the 6th, 12th, 18th, 24th, and 30th floors are the same, and that their coherence coefficients all closely approximate to 1. This phenomenon also proves that there is strong coherence between different floors, and that the vibration modes of different floors are in good agreement with the first-order vibration modes. All the phase angles of cross-spectral density function on the 6th, 12th, and 18th floors are 0° when the frequency of wave crest is 2.3 Hz, and the coherence coefficient is 1, which shows that the vibration directions of the 6th, 12th, and 18th floors are the same, and have high cooperativity. While the phase angle of the cross-spectral density function on the 30th floor is 180°, which shows that the vibration direction of the 30th floor at the second order frequency is opposite to that at the 6th, 12th, and 18th floors. The phase angle of the cross-spectral density function on the 24th floor is within the range of 0~180°, and is nearly 0°, which indicates that the vibration

directions of the 24th floor have a higher vibration cooperativity with the 6th, 12th, and 18th floors than with the 30th floor, and that the vibration modes of different floors are in good agreement with the second-order vibration modes. The phase angles of the cross-spectral density function of the 6th, 12th, and 30th floors are 180° when the frequency of wave crest is 4.2 Hz, and those of the 18th and 24th floors are 0°, which demonstrates that there are consistent vibrations among the 6th, 12th, and 30th floors and between the 18th and 24th floors. While the vibrations existing among the 6th, 12th, and 30th floors are opposite to those existing between the 18th and 24th floors, and the coherence coefficients of the 18th and 24th floors all approach 1, which proves that the vibration cooperativity at those two points is extremely high and that vibration modes of different floors are consistent with the third-order vibration modes. The same principle as above is adopted to analyze the measured cross-spectral density, phase spectrum, and coherence functions along the Y axis and the torsional direction. The measured CSD, PS, and CF of the first three modes along the X and Y axes axis are shown in Table 1. The measured CSD, PS, and CF of the first three modes along the torsional direction are shown in Table 2.

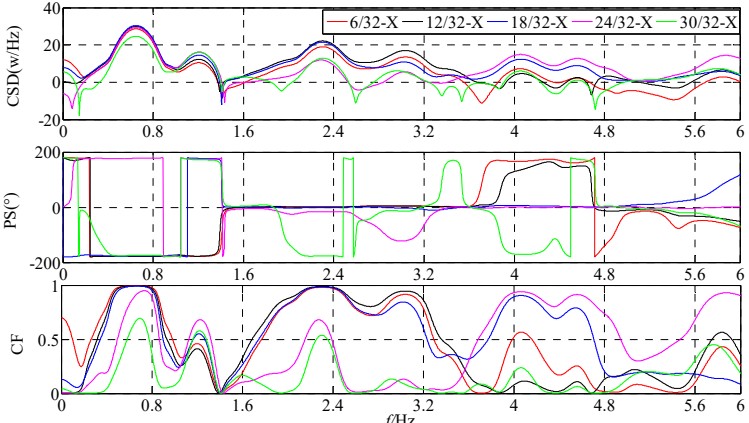

**Figure 11.** The measured CSD, PS, and CF along the X axis.

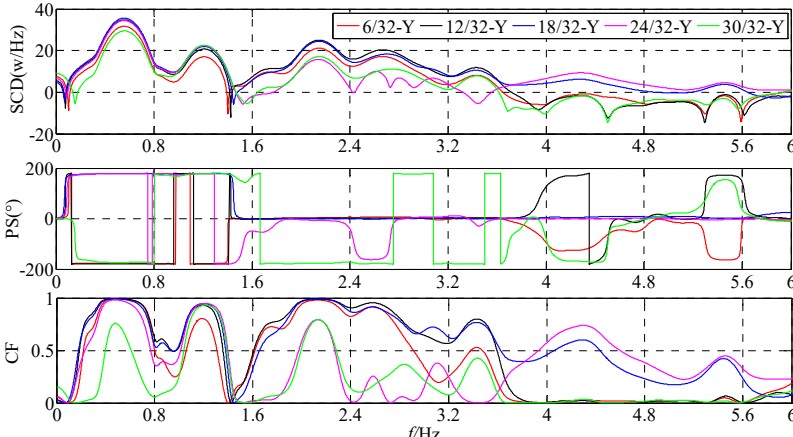

**Figure 12.** The measured CSD, PS, and CF along the Y axis.

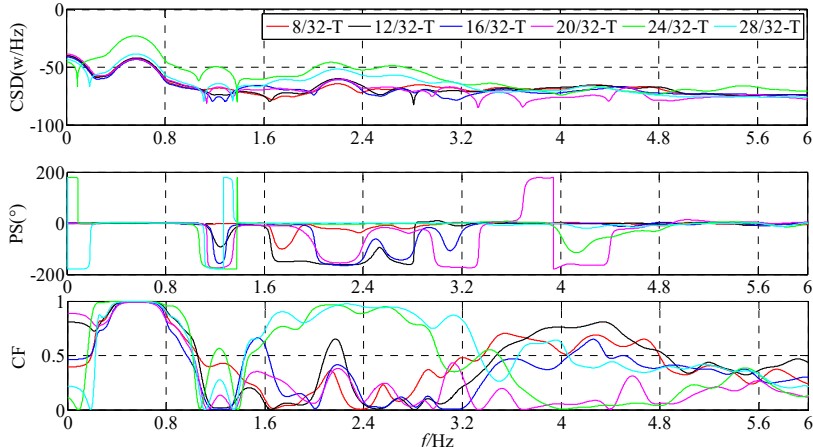

**Figure 13.** The measured CSD, PS, and CF along the torsional direction.

**Table 1.** The measured CSD, PS, and CF of the first three modes along the X and Y axes.

| Mode Number | Relevant Floor | X Axis | | | Y Axis | | |
|---|---|---|---|---|---|---|---|
| | | Natural Frequency by CSD (Hz) | PS | CF | Natural Frequency by CSD (Hz) | PS | CF |
| First-order mode | 32/6 | 0.66 | 180° | 1.00 | 0.56 | 180° | 1.00 |
| | 32/12 | 0.66 | 180° | 1.00 | 0.56 | 180° | 1.00 |
| | 32/18 | 0.66 | 180° | 1.00 | 0.56 | 180° | 1.00 |
| | 32/24 | 0.66 | 180° | 0.90 | 0.56 | 180° | 0.99 |
| | 32/30 | 0.66 | 180° | 0.70 | 0.56 | 180° | 0.75 |
| Second-order mode | 32/6 | 2.30 | 0° | 1.00 | 2.15 | 0° | 1.00 |
| | 32/12 | 2.30 | 0° | 1.00 | 2.15 | 0° | 1.00 |
| | 32/18 | 2.30 | 0° | 1.00 | 2.15 | 0° | 1.00 |
| | 32/24 | 2.30 | 20° | 0.70 | 2.15 | 10° | 0.75 |
| | 32/30 | 2.30 | 180° | 0.55 | 2.15 | 180° | 0.75 |
| Third-order mode | 32/6 | 4.20 | 180° | 0.60 | 4.30 | 150° | 0.05 |
| | 32/12 | 4.20 | 170° | 0.20 | 4.30 | 180° | 0.03 |
| | 32/18 | 4.20 | 5° | 0.90 | 4.30 | 0° | 0.6 |
| | 32/24 | 4.20 | 0° | 0.95 | 4.30 | 0° | 0.75 |
| | 32/30 | 4.20 | 180° | 0.25 | 4.30 | 180° | 0.05 |

**Table 2.** The measured CSD, phase spectrum, and coherence functions of the first three mode along the torsional direction.

| Mode Number | Relevant Floor | Torsional Direction | | |
|---|---|---|---|---|
| | | Natural Frequency by CSD (Hz) | PS | CF |
| First-order mode | 32/8 | 0.56 | 0° | 1.0 |
| | 32/12 | 0.56 | 0° | 1.0 |
| | 32/16 | 0.56 | 0° | 1.0 |
| | 32/20 | 0.56 | 0° | 1.0 |
| | 32/24 | 0.56 | 0° | 1.0 |
| | 32/28 | 0.56 | 0° | 1.0 |
| Second-order mode | 32/8 | 2.15 | 20° | 0.37 |
| | 32/12 | 2.15 | 180° | 0.70 |
| | 32/16 | 2.15 | 180° | 0.42 |
| | 32/20 | 2.15 | 170° | 0.40 |
| | 32/24 | 2.15 | 0° | 0.97 |
| | 32/28 | 2.15 | 0° | 0.97 |
| Third-order mode | 32/8 | 4.30 | 0° | 0.7 |
| | 32/12 | 4.35 | 0° | 0.8 |
| | 32/16 | 4.35 | 5° | 0.65 |
| | 32/20 | 4.35 | 180° | 0.1 |
| | 32/24 | 4.35 | 100° | 0.04 |
| | 32/28 | 4.35 | 25° | 0.4 |

Vibration mode, frequency, and damping ratio are key modal parameters to commonly reflect dynamic characteristics of structures. According to the synthetic evaluation of the auto-power, cross-spectral, phase spectrum, and coherence functions of different floors along the X axis, Y axis, and torsion direction, modal frequencies corresponding to the first three vibration modes can be clearly diagnosed. The first three order frequencies along the X axis were 0.66 Hz, 2.30 Hz, and 4.20 Hz in sequence, and those along the Y axis were 0.56 Hz, 2.15 Hz, and 4.30 Hz, while those along the torsional direction were 0.56 Hz, 2.15 Hz, and 4.35 Hz, respectively. The first three vibration modes along the X axis, Y axis, and the torsional direction of the measured building were calculated based on the parameters of acceleration APSD, CSD, PS, and CF. Besides, the floors from below along the X and Y axes were the 6th, 12th, 18th, 24th, 30th, and 32th floors, while those along the torsional direction were the 8th, 12th, 16th, 20th, 24th, 28th, and 30th floors, and the torsional vibration of the 4th floor approximated half of that in the 8th floor. The measured vibration modes at the first three orders along the X and Y axes and the torsional direction are shown in Figure 14.

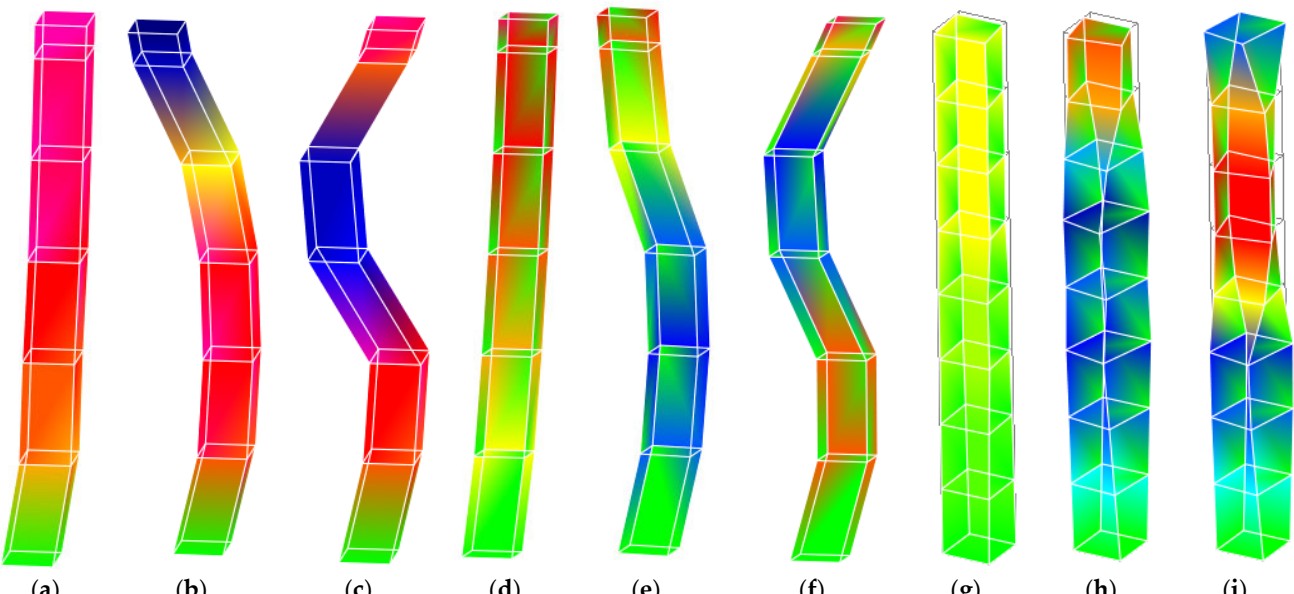

**Figure 14.** The measured vibration modes at the first three orders along the X and Y axis and torsional direction: (**a**) the first-order vibration mode along the X axis; (**b**) the second-order vibration mode along X axis; (**c**) the third-order vibration mode along the X axis; (**d**) the first-order vibration mode along the Y axis; (**e**) the second-order vibration mode along the Y axis; (**f**) the third-order vibration mode along the Y axis; (**g**) the first-order vibration mode along the torsional direction; (**h**) the second-order vibration mode along the torsional direction; and (**i**) the third-order vibration mode along the torsional direction.

## 4. Experimental Modal Parameters Analysis

### 4.1. Comparisons of the Measured Vibration Modes and Those Simulated by Finite Element Method

As shown in Figure 14, the first-order vibration modes of all measured floors along the torsional direction are basically the same. The second-order vibration modes of the 6th, 12th, and 18th floors along the X and Y axes have an opposite vibration direction with that of the 30th floor, while the third-order vibration modes of the 6th, 12th, and 30th floors along the X and Y axes have an opposite direction to those of the 18th and 24th floors. Besides, along the torsional direction, the vibration directions of the 24th and 28th floors were opposite to those of the 8th, 12th, and 16th floors at the second-order frequencies, and the vibration directions of the 8th, 12th, 16th, and 28th floors were opposite to those of the 20th and 24th floors. All these results were all in good agreement with those identified by the measured phase spectrum, which indicated that there were few disturbances when measuring the responses of each floor, thus the inherent attributes of the highrise building can be also recognized. Meanwhile, modal assurance criteria (MAC) were adopted to check reliability, and its range was 0~1. If the value of MAC was more than 0.7, it meant that there was huge correlation between the two order modes, and that there may exist false modes or unknown branch structures. The verification results of modal correlation along the X axis, Y axis, and torsional direction are shown in Figure 15.

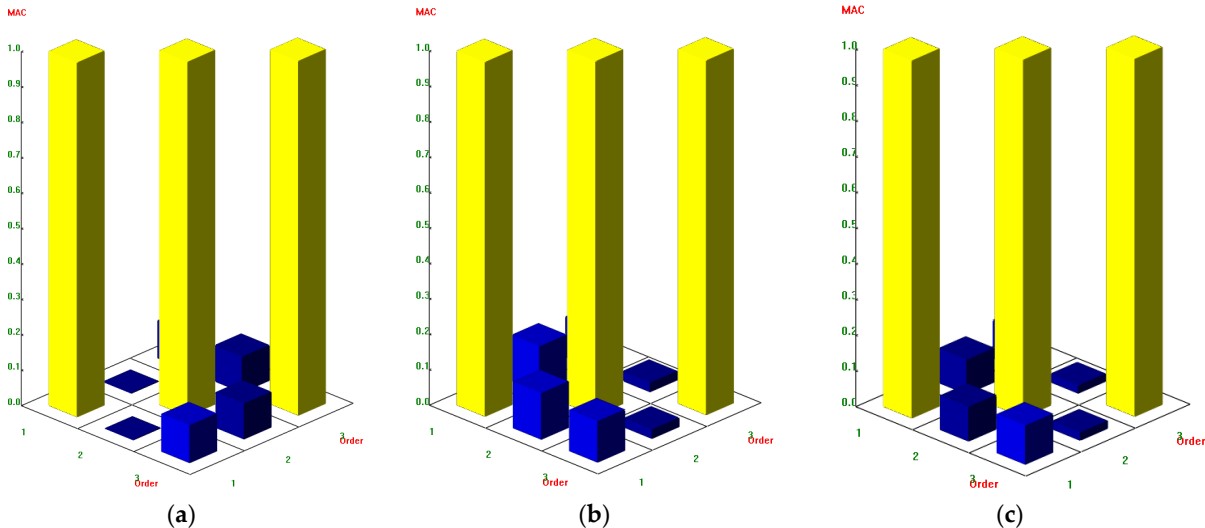

**Figure 15.** Correlation verification of modes at the first three orders along the X axis, Y axis and torsional direction: (**a**) correlation verification of modes at the first three orders along the X axis; (**b**) correlation verification of modes at the first three orders along the Y axis; (**c**) correlation verification of modes at the first three orders along the torsional direction.

According to Figure 15, MAC of the first three order vibration modes along the X axis, Y axis, and the torsional direction are all less than 0.2, which presents that there are weak correlations between different order modes, and that the first three-order modal vibration modes are reliable. Meanwhile, finite element numerical analysis (FENA) Midas/Gen was also applied to conduct the numerical simulation and Eigenvalue calculations of the high-rise building, and the major structure was simulated for the beam and plate element. The equivalent mass of filler wall, glass curtain wall, and other similar structures was calculated without considering stiffness, and these structures were all loaded linearly on the beam elements. Because the filler wall has windows and doors, a constant load conversion coefficient of 0.75 was adopted to conduct the numerical simulation. The top side of building foundation was regarded as the fixed end, and only the load transfer of foundation and superstructure was considered in the process of numerical simulation regardless of the interrelationship between foundation and superstructure and that between the foundation soil and building within basement range. Figure 16 presents the finite numerical simulation results of the first three-order vibration modes along the X axis, Y axis, and torsional directions. Table 3 shows the comparison of the measured values and those calculated by definite element numerical simulation of the first three-order frequencies, and the comparison results show that the simulation results of the first three-order vibration modes along X axis, Y axis, and torsional direction are in good accordance with those of the measured results, and that the deviations of the simulated modal frequencies are all within 7%.

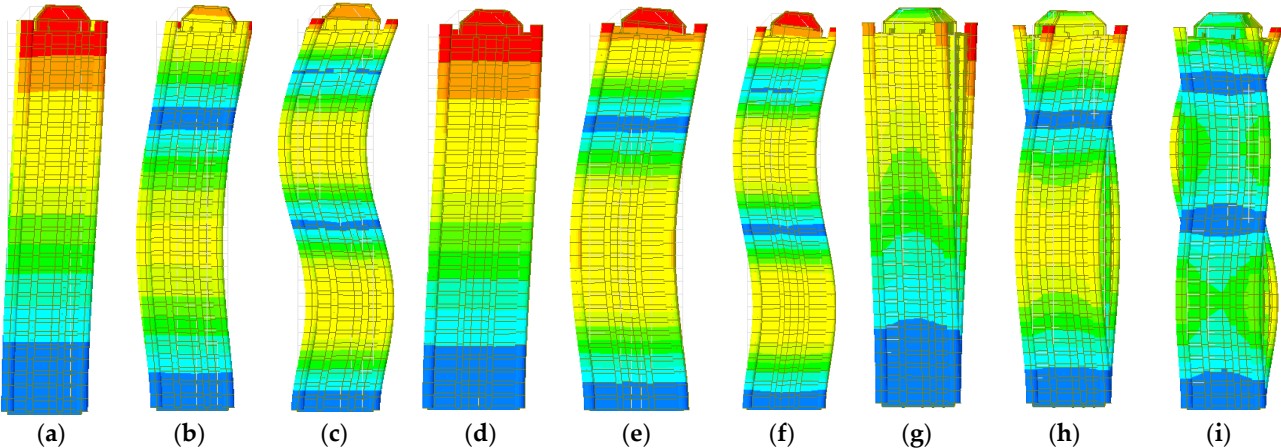

(a)  (b)  (c)  (d)  (e)  (f)  (g)  (h)  (i)

**Figure 16.** The finite numerical simulation results of the first three-order vibration modes along the X and Y axes and torsional direction: (**a**) the finite numerical simulation results of the first-order vibration mode along the X axis; (**b**) the finite numerical simulation results of the second-order vibration mode along X axis; (**c**) the finite numerical simulation results of the third-order vibration mode along the X axis; (**d**) the finite numerical simulation results of the first-order vibration mode along the Y axis; (**e**) the finite numerical simulation results of the second-order vibration mode along the Y axis; (**f**) the finite numerical simulation results of the third-order vibration mode along the Y axis; (**g**) the finite numerical simulation results of the first-order vibration mode along torsional direction; (**h**) the finite numerical simulation results of the second-order vibration mode along torsional direction; and (**i**) the finite numerical simulation results of the third-order vibration mode along torsional direction.

**Table 3.** Comparisons of the finite numerical simulation values and the measured values of the first three-order modes and frequencies along the X axis, Y axis, and torsional direction.

| Modal Number | Mode | | Natural Frequency/Hz | | |
| --- | --- | --- | --- | --- | --- |
| | Measured | Finite Element Simulation | Measured | Finite Element Simulation | Relative Deviation (%) |
| 1 | The first-order translational mode along the Y axis | The first-order translational mode along the Y axis | 0.56 | 0.56 | 0.00 |
| 2 | The first-order torsional mode | The first-order torsional mode | 0.56 | 0.60 | 6.67 |
| 3 | The first-order translational mode along the X axis | The first-order translational mode along the X axis | 0.66 | 0.66 | 0.00 |
| 4 | The second-order translational mode along the Y axis | The second-order translational mode along the Y axis | 2.15 | 2.07 | 3.86 |
| 5 | The second-order torsional mode | The second-order torsional mode | 2.15 | 2.10 | 2.38 |
| 6 | The second-order translational mode along the X axis | The second-order translational mode along the X axis | 2.30 | 2.24 | 2.68 |
| 7 | The third-order translational mode along the X axis | The third-order translational mode along the X axis | 4.20 | 4.48 | 6.25 |
| 8 | The third-order translational mode along the Y axis | The third-order translational mode along the Y axis | 4.30 | 4.53 | 5.08 |
| 9 | The third-order torsional mode | The third-order torsional mode | 4.35 | 4.63 | 6.05 |

*4.2. Measured Modal Frequency under Different Wind Speeds*

4.2.1. Identification Method of Modal Parameters

The amplitude of signal interception and the collective average superposition times are vital in adopting the random decrement technique (RDT) to obtain feature recognition. If the amplitude of the signal interception is too small, the superposition effect will be undesirable, though the collective average superposition times can be enlarged. If the amplitude of signal interception is too large, the corresponding superposition times of the sub-signals will be reduced, and the negative correlation between the amplitude of signal interception and superposition times will arise as well. To obtain the favorable random decrement signals, this paper intercepted both the positive and negative amplitudes at the same time, then modified different interception amplitudes in accordance with the reference interception amplitude, and lastly assembled and superposed all sub-signals at the average interception amplitude in accordance with the reference interception amplitude. Generally speaking, the interception amplitude is 1.5 times of the standard deviation of the input signal (1.5 S). Namely, the reference interception amplitude was defined as 1.5 times the standard deviation (1.5 S), the interval of interception amplitude was $\tau$, and the range of interception amplitude $k_j$ was 1.0~2.0 times the standard deviation (1.0~2.0 S). The sub-signals were attained from those signals with basic time intervals of 10 min, then the intercepted amplitude was modified as 1.5 times the standard deviation, and all the sub-signals were superposed and equally divided in the end, thus the favorable random decrement signals were obtained. Concrete operation procedures are shown as follows:

$$s = 1.5\sqrt{\frac{1}{n-1}\sum_{i=1}^{n} x_i - \overline{X}} \tag{10}$$

$$x_{k_j}(t)^+ = \frac{\sum_{j=1}^{j=m}(\frac{1.5}{k_j}x(t_{k_j})^+)}{m(k_j)} \quad (k_{j=1} = s/1.5, \ k_{j+1} = k_j + \tau, \ \tau = 1/m) \tag{11}$$

$$x_{k_j}(t)^- = \frac{\sum_{j=1}^{j=m}(\frac{1.5}{k_j}x(t_{k_j})^-)}{m(k_j)} \tag{12}$$

where $x_i$ is the sample of discrete experimental data, $n$ is the number of samples, $\overline{X}$ refers to the mean value of sub-samples, $x(t_{k_j+\tau})^+$ represents the free attenuation curves by intercepting the positive amplitude sample, $x(t_{k_j+\tau})^-$ stands for the free attenuation curves by intercepting the negative amplitude sample, $\tau$ denotes the interval of interception amplitude, $m$ stands for the number of interception amplitude sample, and $k_j$ refers to the amplitude coefficient at the $j$th time. $1.5/k_j$ is the modification coefficient of interception amplitude at the corresponding 1.5 times of the reference amplitude.

The superimposed free attenuation curve can be written as

$$x_{k_j}(t) = \frac{x_{k_j}(t)^+ - x_{k_j}(t)^-}{2} \tag{13}$$

For Equations (10)–(13), if the interception range is within the 1.0~2.0 times the standard deviation, the interval of interception amplitude $\tau$ is equivalent to 0.1 times of the standard deviation, thus the superposition times during the whole process have increased by 40 times compared with pre-improvement. In this condition, the superposition times are completely capable of meeting the RDT requirements when the basic time duration of 10 min is taken into consideration. After obtaining the free vibration attenuation curve, the logarithmic decrement method was used, and the peak values of the free attenuation curve were attained to calculate damping parameters. Moreover, the fitting formula can be expressed as follows:

$$y(t) = a \cdot \exp(-\zeta\omega t) \qquad (14)$$

where $a$ is the amplitude of acceleration response when $t$ equals 0, namely $a = y(0)$; $\zeta$ denotes the structural damping ratio; and $\omega$ is the circular frequency. This paper firstly sets reasonable band-pass filtering for the first three-order frequencies of the measured data, and the band-pass range of filter along the X and Y axes was determined as $f \in (f_i - 0.08 f_i, f_i + 0.08 f_i)$, where $f_i$ is the frequency identified by spectrum peaks with the time duration of 10 min, it can be separating out the first three-order frequency resonance signals in the high-rise building, and then the frequency $f$ was identified according to the free vibration attenuation curve, whereas the circumference ratio was calculated by the formula $\omega = 2\pi f$. Figure 17 is the first-order free attenuation curve of several samples calculated by RDT along the transnational and torsional directions when the basic time duration is 10 min.

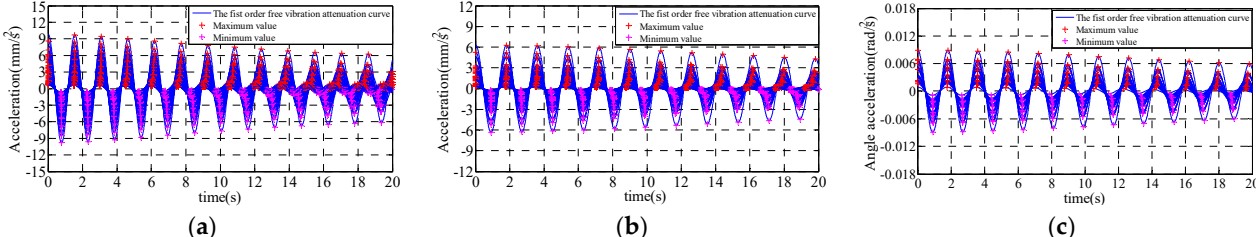

**Figure 17.** The first-order free vibration attenuation curve along the translational and torsional directions identified by RDT: (**a**) free vibration attenuation curve along the X axis; (**b**) free vibration attenuation curve along the Y axis; (**c**) free vibration attenuation curve along the torsional direction.

4.2.2. Calculation Method of Fundamental Frequencies of High-Rise Buildings under Different Wind Speeds

The natural frequency of a high-rise building is the natural attribute of structural vibration. The natural frequency is an important index to judge whether the structural stiffness, quality, and stiffness are reasonable. The existing codes estimate the fundamental frequency of high-rise buildings, but they do not evaluate the impact of a strong typhoon on the fundamental frequency of high-rise buildings. According to the measured data of natural frequency of super high-rise buildings under different wind speeds, the most direct correlation factor causing the change in modal parameters of high-rise buildings is the wind speed. The samples were selected from all the measured data of typhoon Sarika. Because the coupling effects of structures would not remain the same in different wind speeds, the first three-order modal frequencies along the X axis, Y axis, and torsional direction under different mean wind speed were obtained based on the sufficient experimental data to analyze the relationship between the first three-order modal frequencies and wind speed. It can be known from Figure 18 that the first three-order modal frequencies along the X and Y axes and the torsional direction decrease linearly as the mean wind speed increases, which explains that the coupling effects between wind and structures are more obvious as wind speed increases, and those effects have great influences on the

modal frequencies of structures, thus needing to be emphasized in wind-resistant designing of high-rise buildings. It also can be concluded from the comparisons of frequencies along the Y axis and those along the torsional direction shown in Figure 18 that their sizes and development trends are almost the same. Therefore, the frequencies in the torsional direction of high-rise buildings were close to those along the Y axis.

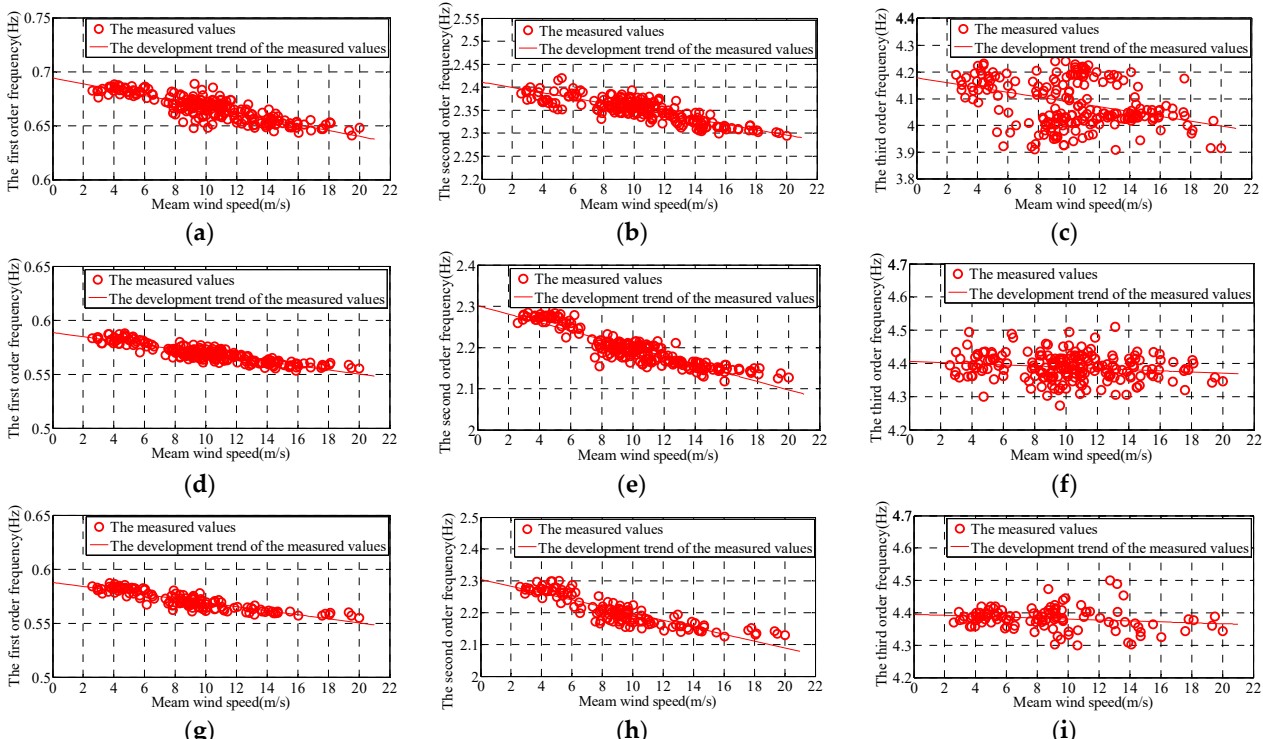

**Figure 18.** The relationships between the first three-order modal frequencies and mean wind speeds along the X axis, Y axis, and torsional direction: (**a**) The first-order modal frequency along the X axis; (**b**) The second-order modal damping ratio along the X axis; (**c**) The third-order modal damping ratio along the X axis; (**d**) The first-order modal frequency along the Y axis; (**e**) The second-order modal frequency along the Y axis; (**f**) The third-order modal frequency along the Y axis; (**g**) The first-order modal frequency along the torsional direction; (**h**) The second-order modal frequency along the torsional direction; (**i**) The third-order modal frequency along the torsional direction.

The fundamental natural period, which is the important standard to judge the reliability of the stiffness and to evaluate whether mass matches with the stiffness of structures, is the inherent basic dynamic characteristic of high-rise buildings. The above analysis shows that wind load has great influences on the fundamental frequencies of high-rise buildings, while codes in many countries (American UBC 1997 [30], European codes [31], Japanese Building Standards Law [32], and Load Codes of China [33]) have not taken the influences of wind-induced response on the fundamental frequencies into consideration, thus the fundamental frequencies calculation formulas of high-rise buildings considering the influences of wind speed were put forward in this paper:

$$\frac{1}{T_{(l,s,t)}(\overline{V})} = \frac{1}{T_{0(l,s,t)}} - \alpha_{(l,s,t)}\overline{V} \tag{15}$$

In Equation (15), $T_l$、$T_s$、$T_t$ represent the fundamental natural periods along the X axis, Y axis, and torsional direction, respectively, when the basic time duration of

the mean wind speed($\overline{V}$ m/s) is 10 min. $T_{0l}$、$T_{0s}$、$T_{0t}$ represent the fundamental natural periods along the X axis, Y axis, and torsional direction under static wind condition. $\overline{V}$ is the mean wind speed within 0~20 m/s when the basic time duration is 10 min; H is the total height of the building. $\alpha_{(l,s,t)}$ is the wind load sensitivity coefficient of fundamental frequency, and the corresponding $\alpha_{(l,s,t)}$ of the measured high-rise building along the X axis, Y axis, and the torsional direction are 0.0027, 0.0019, and 0.0019 respectively.

## 5. Conclusions

The measured results of the translational and torsional acceleration responses of the high-rise building in Haikou under the influences of typhoon Sarika were analyzed in detail in this paper, by which the vibration modes, natural frequencies, and damping ratios were obtained, and the variation laws of the first three-order frequencies and modal damping ratios were also investigated. On the basis of the measured data, the conclusions below are drawn:

(1) The measured value of the maximum instantaneous wind speed atop the building is 34.84 m/s, and that of mean wind speed is about 20.02 m/s when the basic time duration is 10 min. Besides, the peak value of the maximum angular acceleration at the top floor along the torsional direction is 0.031 rad/s$^2$.

(2) The auto-power spectral density function, cross-spectral density function, phase spectrum, and coherence function can be applied to identify the first three-order modal frequencies of the building and to judge where different floors locate in the vibration shapes of resonance region. On the basis of phase spectrum and coherence functions, it can be judged whether the corresponding frequencies of the auto-power spectral and cross-spectral density functions at wave crests are true modal frequencies.

(3) The first three-order vibration modal frequencies decrease linearly with the growth in mean wind speed when the mean wind speed is within the range of 0~20 m/s, which indicates that wind-induced vibration has significant influences on the high-rise buildings.

(4) The existing codes can only estimate the fundamental frequency of high-rise buildings under static wind, but the frequency estimation of high-rise buildings under typhoon is not involved. This paper took the influences of wind-induced response on fundamental frequencies into consideration under the action of typhoons, and then the estimation formula of the modal frequencies of high-rise buildings considering the influences of different wind speeds was put forward, the formula can calculate the natural vibration frequency of high-rise buildings under strong wind in the design phase in a precise way.

**Author Contributions:** J.H. contributed to the overall study design, analysis, and writing of the manuscript. Z.L. provided technical support and supervision. Z.Z. provided technical support and modification-polis. All authors have read and agreed to the published version of the manuscript.

**Funding:** This research was financially supported by the Natural Science Foundation of Hunan Province, China (Grant No. 2020JJ5205), Scientific Research Project of Hunan Provincial Department of Education, China (Grant No. 21B0730) and the National Natural Science Foundation of China (Grant No. 91215302).

**Institutional Review Board Statement:** Not applicable.

**Informed Consent Statement:** Not applicable.

**Conflicts of Interest:** The authors declare no conflict of interest.

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
