# Peer review of "Estimation Formula of Modal Frequency of High-Rise Buildings under Different Wind Speeds during Typhoons"

_applsci, doi:10.3390/app12010047_

Round 1
Reviewer 1 Report
This paper provides useful information on wind response of high-rise buildings,
but the English language is poor.
Please correct the following points and improve the English in general.
1) There are no floor plans or elevations of the building except the photograph.
The information about the building must be provided.
2) Fig. 1(b) is labeled as "Plan view", but it is actually a photograph.
3) In Page 4, line 117, "height" is probably a mistake for "weight".
4) I don't understand the meaning of the symbol for s2θ in Fig. 2.
5) Fig. 4 needs a Z axis.
6) In Page 6, line 142, the position of the explanation of m1 and m2 is wrong.
7) There is no explanation for Fig. 5.
8) In Page 7, line 167 and 168, the same sentence "under the influences of typhoon Sarika" is repeated.
9) In Page 9, line 225, Fig. 12 is probably a mistake for Fig. 9.
10) I don't understand the meaning of Page 9, line 229, "the measured Fig. of vibration mode".
11) In Page 10, line 237-246, you mention the effect of building height,
but there is no evidence.
12) In Page 11, line 277, Fig. 13, Fig. 14, and Fig. 15 are probably wrong for
Fig. 10, Fig. 11, and Fig. 12.
13) In Page 16, line 393, "Finite Element Mumerial Analysis" is probably a mistake for "Finite Element Numerical Analysis".
14) In Fig. 15, the aspect ratio is not drawn correctly in the building diagram
for each mode.
15) In Page 22, line 526, what is the calculation method in the European codes? I could not find the equation as Equation (15) in Eurocode.
Author Response
This paper provides useful information on wind response of high-rise buildings, but the English language is poor. Please correct the following points and improve the English in general.
Comment: 1) There are no floor plans or elevations of the building except the photograph. The information about the building must be provided.
Response: Thank you for the comment. We added the Building plan of the measured high-rise building, see Figure 2 for details.
Comment: 2) Fig. 1(b) is labeled as "Plan view", but it is actually a photograph.
Response: Thank you for the comment. We corrected the labeled illustration of Fig. 1(b) , see Figure 1(b) for details.
Comment: 3) In Page 4, line 117, "height" is probably a mistake for "weight".
Response: Thank you for the comment. We modified the sentence in line 120.
“Model RM.Young 05103V mechanical anemometer was used to collect the wind field and was installed at the vertical weight of 115m.”
Comment: 4) I don't understand the meaning of the symbol for s2θ in Fig. 2.
Response: Thank you for the comment. According to fig. 6, the symbol for s2θ Meaning "Ground rotation acceleration".
Comment: 5) Fig. 4 needs a Z axis.
Response: Thank you for the comment. We added the Z axis to Figure 5.
Comment: 6) In Page 6, line 142, the position of the explanation of m1 and m2 is wrong.
Response: Thank you for the comment. We corrected the position of m1 and m2 ( See Line 158-159 ).
Comment: 7) There is no explanation for Fig. 5.
Response: Thank you for the comment. We added a description to Figure 6 as follows:
Mathematical model of the strong earthquake rotation accelerometer
“The mathematical representation of the mathematical model of the rotation accelerometer is shown in Figure 6.”
Comment: 8) In Page 7, line 167 and 168, the same sentence "under the influences of typhoon Sarika" is repeated.
Response: Thank you for the comment. We deleted the repeated sentence in the corresponding place.
Comment: 9) In Page 9, line 225, Fig. 12 is probably a mistake for Fig. 9.
Response: Thank you for the comment. We corrected the descriptive error in the manuscript In line 225.
“It can be known from Fig. 10 that the auto-power spectral density functions along X, Y axis”
Comment: 10) I don't understand the meaning of Page 9, line 229, "the measured Fig. of vibration mode".
Response: Thank you for the comment. We corrected the descriptive error in the manuscript In line 237.
“which can be verified in the measured torsional vibration mode of Fig.14(g).”
Comment: 11) In Page 10, line 237-246, you mention the effect of building height, but there is no evidence.
Response: Thank you for the comment. I agree with you. In order to avoid the influence of subjective factors, we have deleted the inappropriate statement.
“Because the wind energy is concentrated in the low-frequency region, it has a great impact on the measured low-order modal frequency of high-rise buildings. The measured results show that the frequency value at the self-spectral peak is significantly less than that at the low wind speed peak in the horizontal and torsional turning high-speed stage, indicating that the influence of the vibration amplitude of high-rise buildings on the horizontal and torsional turning modal parameters can not be ignored.”
Comment: 12) In Page 11, line 277, Fig. 13, Fig. 14, and Fig. 15 are probably wrong for Fig. 10, Fig. 11, and Fig. 12.
Response: Thank you for the comment. We corrected the descriptive errors in the manuscript In line 284.
“More details are presented in Figs. 11-13.”
Comment: 13) In Page 16, line 393, "Finite Element Mumerial Analysis" is probably a mistake for "Finite Element Numerical Analysis".
Response: Thank you for the comment. We have modified "final element numerical analysis" to "final element numerical analysis" in the manuscript.
“the Finite Element Numerial Analysis (FENA) microsoft Midas Gen was also applied to conduct numerial simulation and eigenvalue calculations of the high-rise building”
Comment: 14) In Fig. 15, the aspect ratio is not drawn correctly in the building diagram for each mode.
Response:Thank you for the comment. In Fig. 16, we adjusted the aspect ratio for each mode.
Comment: 15) In Page 22, line 526, what is the calculation method in the European codes? I could not find the equation as Equation (15) in Eurocode.
Response: Thank you for the comment. Our inappropriate statement should be responsible for your misunderstanding, and we have corrected. Equation (15) is a calculation formula based on the analysis results of measured data of super high-rise buildings under the action of typhoon. At present, the existing codes can only estimate the modal frequency of high-rise buildings under static wind.

Reviewer 2 Report
The manuscript has merits and the topic is relevant, however, some problems have to be addressed and the paper has to be revised. See my comments below:
- English.
- Please, check the typological errors (e.g., 'numerial' should be numerical; 'vibraiton' should be vibration; 'angel' should be angle; 'root mean squares' should be the root mean square; 'wind filed' should be wind field...). Is it 'rotation accelerometer' or a 'rotational accelerometer'?
- Check the grammar.
- Write shorter, but clearer sentences. Sometimes the meaning is lost (e.g., Line 118, Lines 165-168, Line 229; Lines 259-260; Lines 481-484). - Abstract. Add the most important findings to the text of the Abstract.
- Methods. This needs improving. I recommend presenting this part of the paper, especially pages 10-15 in a more concise way. Try to use less text and more Tables. Try adding clearer subsections.
- Other corrections:
- Check the Template, as it seems that the Figure captions are too large
- Fig. 3: try improving the DPI resolution of the text in the image
- Line 146: the equation is not numbered, give it a number
- Line 156: the time is written in the 24-hour format and in Line 94, it is in 12-h format. Please, unify the format.
- Line 181: correct the 0.079 m/s2
- Line 206: periodic diagrams were used, not was used
- Line 245: related to, not related with
- Line 293: (they) were in good agreement, not in well agreement
- Line 323: ...all approximated to 1 - is this correctly phrased?
- Fig. 13: Please, add the floor numbers to the picture for a better orientation
- Line 386: 'along X-axis' is written twice in a row
- Line 393: Microsoft (capital M) + maybe add a link to the software
- Line 395: simulated for, not by
- Lines 515-524: This is the part that should be in the Discussion or the Conclusion.
- Line 528: Is the equation necessary at the end of the paper? Cannot it be placed elsewhere?
5. Conclusion/Results. I recommend extending this section and describing the application of the presented method in practice - in which phase of designing, using which software... This is the most important contribution of the paper.
Author Response
The manuscript has merits and the topic is relevant, however, some problems have to be addressed and the paper has to be revised. See my comments below:
Comment: 1.English.
(1)- Please, check the typological errors (e.g., 'numerial' should be numerical; 'vibraiton' should be vibration; 'angel' should be angle; 'root mean squares' should be the root mean square; 'wind filed' should be wind field...). Is it 'rotation accelerometer' or a 'rotational accelerometer'?
(2)- Check the grammar.
(3)- Write shorter, but clearer sentences. Sometimes the meaning is lost (e.g., Line 118, Lines 165-168, Line 229; Lines 259-260; Lines 481-484).
Response:
(1)Thank you for the comment and careful check. We have checked the typological errors carefully and corrected them one by one to improve our manuscript.
(2)Thank you for the comment. We invited a professional English teacher to check the grammar, and the corresponding changes are highlighted under the revision mode.
(3)Thank you for the comment. I agree with you. We have modified the corresponding statements and added the necessary instructions.
Comment: 2.Abstract. Add the most important findings to the text of the Abstract.
Response: Thank you for the comment. We revised the text of the abstract and added the most important findings to the Abstract.
Comment: 3.Methods. This needs improving. I recommend presenting this part of the paper, especially pages 10-15 in a more concise way. Try to use less text and more Tables. Try adding clearer subsections.
Response: Thank you for the comment. Thank you for your valuable suggestions. We have made the corresponding adjustments on page 14-16 of the article.
Comment: 4.Other corrections:
Check the Template, as it seems that the Figure captions are too large
Fig. 3: try improving the DPI resolution of the text in the image
Line 146: the equation is not numbered, give it a number
Line 156: the time is written in the 24-hour format and in Line 94, it is in 12-h format. Please, unify the format.
Line 181: correct the 0.079 m/s2
Line 206: periodic diagrams were used, not was used
Line 245: related to, not related with
Line 293: (they) were in good agreement, not in well agreement
Line 323: ...all approximated to 1 - is this correctly phrased?
Fig. 13: Please, add the floor numbers to the picture for a better orientation
Line 386: 'along X-axis' is written twice in a row
Line 393: Microsoft (capital M) + maybe add a link to the software
Line 395: simulated for, not by
Lines 515-524: This is the part that should be in the Discussion or the Conclusion.
Line 528: Is the equation necessary at the end of the paper? Cannot it be placed elsewhere?
Response: Thank you for your comments. Those comments are all valuable and very helpful in revising and improving the paper, as well as in providing guidance significance to the researches. The comments have been studied carefully and the corresponding changes have been highlighted under the revision mode.
Comment: 5. Conclusion/Results. I recommend extending this section and describing the application of the presented method in practice - in which phase of designing, using which software... This is the most important contribution of the paper.
Response: Thank you for your comments. The existing codes can only estimate the fundamental frequency of high-rise buildings under static wind, but the frequency estimation of high-rise buildings under typhoon is not involved, This paper taken the influences of wind load on fundamental frequencies into considerations, and then the estimation formula of the modal frequencies of high-rise buildings considering the influences of different wind speeds was put forward, the formula can more accurately calculate the natural vibration frequency of high-rise buildings under strong wind in the design phase.
The revised manuscript had been modified and made some changes according to your comments.
We appreciate for the editors and reviewers’ warm work earnestly, and hope that the modifications will meet with approval.

Round 2
Reviewer 1 Report
Please correct the title of Fig.1 as below:
Fig. 1. Moving track of typhoon Sarika and the plan view of the measured building
-->
Fig. 1. Moving track of typhoon Sarika and the exterior view of the measured building
Author Response
Comment:Please correct the title of Fig.1 as below:
Fig. 1. Moving track of typhoon Sarika and the plan view of the measured building
-->
Fig. 1. Moving track of typhoon Sarika and the exterior view of the measured building
Response: Thank you for the comment. We corrected the labeled illustration of in Fig. 1.
“FIGURE 1. Moving track of typhoon Sarika and the exterior view of the measured building”
